# When Reality Does Not Meet Expectations—Experiences and Perceived Attitudes of Dutch Stakeholders Regarding Payment and Reimbursement Models for High-Priced Hospital Drugs

**DOI:** 10.3390/ijerph20010340

**Published:** 2022-12-26

**Authors:** Marcelien H. E. Callenbach, Rick A. Vreman, Aukje K. Mantel-Teeuwisse, Wim G. Goettsch

**Affiliations:** 1Division of Pharmacoepidemiology & Clinical Pharmacology, Utrecht Institute for Pharmaceutical Sciences (UIPS), Utrecht University, 3584 CG Utrecht, The Netherlands; 2National Health Care Institute (ZIN), 1112 ZA Diemen, The Netherlands

**Keywords:** managed entry agreements, outcome-based reimbursement, delayed payment models, stakeholder perspectives, pharmaceutical reimbursement, health technology assessment

## Abstract

This study aimed to identify the current experiences with and future preferences for payment and reimbursement models for high-priced hospital therapies in the Netherlands, where the main barriers lie and assess how policy structures facilitate these models. A questionnaire was sent out to Dutch stakeholders (in)directly involved in payment and reimbursement agreements. The survey contained statements assessed with Likert scales, rankings and open questions. The results were analyzed using descriptive statistics. Thirty-nine stakeholders (out of 100) (in)directly involved with reimbursement decision-making completed the survey. Our inquiry showed that currently financial-based reimbursement models are applied most, especially discounts were perceived best due to their simplicity. For the future, outcome-based reimbursement models were preferred, particularly pay-for-outcome models. The main stated challenge for implementation was generating evidence in practice. According to the respondents, upfront payments are currently implemented most often, whereas delayed payment models are preferred to be applied more frequently in the future. Particularly payment-at-outcome-achieved models are preferred; however, they were stated as administratively challenging to arrange. The respondents were moderately satisfied with the payment and reimbursement system in the Netherlands, arguing that the transparency of the final agreements and mutual trust could be improved. These insights can provide stakeholders with future direction when negotiating and implementing innovative reimbursement and payment models. Attention should be paid to the main barriers that are currently perceived as hindering a more frequent implementation of the preferred models and how national policy structures can facilitate a successful implementation.

## 1. Introduction

Over the years medicine expenditures in European countries have been rising substantially [1,2]. The increased introduction of high-priced health interventions poses challenges for healthcare payers and decision-makers, given that healthcare budgets are finite [1,2,3]. Additionally, national competent authorities responsible for pricing and reimbursement (NCAPR), such as healthcare payers and health technology assessment (HTA) organizations, are increasingly faced with uncertainties when a new drug enters the market. This is mainly due to the introduction of therapies that have not been studied comparatively in large populations for longer periods of follow-up at the time of approval, such as those approved through facilitated expedited regulatory pathways, orphan drugs, and advanced therapy medicinal products (ATMPs) [4,5,6,7]. These uncertainties can make decision-making about payment and reimbursement challenging.

The current focus of NCAPR lies in arrangements with marketing authorization holders (hereafter called manufactures) with the aim of sharing the financial risks as well as the responsibility of further evidence generation to diminish uncertainty regarding relative (cost-)effectiveness surrounding the introduction of new technologies or it budget impact [7,8,9,10,11,12,13,14,15,16]. This has led to the development of innovative payment and reimbursement agreements that go beyond an upfront price-setting mechanism, which are often known as managed entry agreements (MEA) or risk-sharing arrangements [7,17,18,19,20]. MEAs have been defined by the OECD as ‘any agreement beyond a yes/no decision on reimbursement between the manufacturer of a therapy and a healthcare payer’ [7]. MEAs include a variety of reimbursement and payment models under specified conditions to address uncertainty about the performance of technologies or to manage the adoption of technologies to maximize their effective use or limit their budget impact [7,17,18,19,20,21,22]. These payment and reimbursement agreements can be categorized in multiple ways. Reimbursement models are often arranged into purely financial (e.g., discounts) and outcome-based agreements (e.g., pay-for-performance) [7,18,19,21,23]. Payment models can be broken down into upfront payments or delayed payment models (e.g., annuity payment and payment at outcomes achieved) [7,18,19,21,23]. Nonetheless, even though countries seem to be using innovative payment and reimbursement models more and more, previous research has shown it remains questionable whether they can deliver on their promises in practice [7,24,25,26].

In the previous literature, stakeholders emphasized that several goals have been achieved by implementing more complex payment and reimbursement models, such as providing earlier patient access to new treatments, reducing remaining (clinical) uncertainties or managing high budget impact [4,7,24]. Nevertheless, it is argued that these agreements are often not transparent and are complex to implement [22,24,27,28]. Additionally, many examples of unsuccessful implementation also exist [29,30,31,32,33]. The extent to which existing and novel policy structures facilitate different payment and reimbursement models seem to be an important factor for their success [7,15,22,24,34]. Additionally, the literature suggests that insight into the experiences and interests of stakeholders involved in making payment and reimbursement arrangements is of great value and can play an essential part in the success of the implementation of innovative reimbursement and payment models [4,7,21,35].

Consequently, there have been several studies on stakeholders’ attitudes toward current and innovative pharmaceutical payment and reimbursement agreements throughout Europe [17,27,36,37]. Previous studies often highlighted the views of one or a few specific stakeholder group(s), most commonly regulators, HTA bodies, payers or manufacturers [17,27,36,37,38,39]. Additionally, previous literature often focused on making a comparison across countries [17,37,40,41]. However, to make novel payment and reimbursement models a success in any single country, insights from all relevant stakeholders including also patient societies, academics, consultancy companies, and healthcare professionals are necessary, specifically within that jurisdiction.

The overall aim of this study was therefore to analyse Dutch stakeholders’ experiences and expectations with payment and reimbursement mechanisms. The specific objectives were to (1) identify the current experiences with and future preferences for payment and reimbursement models for high-priced hospital drugs and their main barriers, (2) assess what the main arguments are for using these arrangements and how current policy structures facilitate them, (3) assess how stakeholders estimate the feasibility of future, more innovative, arrangements within the Dutch policy setting and what in their opinion should change to enable their implementation.

## 2. Methods

### 2.1. Participants

In the Dutch healthcare system, multiple stakeholder groups are involved in payment and reimbursement decision-making. The approached stakeholders were categorized into eleven different groups, which included stakeholders who are directly involved in payment and reimbursement decision-making, but also stakeholders who play a more advisory role or represent the interests of a specific group. These stakeholder groups were: Ministry of Health (1), the National Health Care Institute (Zorginsituut Nederland, ZIN) (2), Pharmaceutical industry (3), Healthcare insurers (4), Pooled procurement organizations (5), Medical specialist societies (6), Private/academic hospitals (7), Patients organizations (8), Consultancy (9), Academia (10) and Others (11) such as interest groups (e.g., medical specialist societies and patient organization). Invited stakeholder representatives were selected based on their seniority and involvement in reimbursement and payment models in the Netherlands. Contacts were gathered through contacts of the authors and purposeful sampling [42]. When analysing the results, these stakeholder groups where further clustered into eight main perspectives, namely a ministerial perspective, HTA perspective, healthcare payer perspective, healthcare provider perspective, a pharmaceutical industry perspective, healthcare consultant perspective, an academic perspective, and an interest group perspective (Appendix A contains an overview of which stakeholder groups were categorized under which perspective).

### 2.2. Survey Objective and Design

Information about the experiences of stakeholders with different payment and reimbursement models was collected through a survey. The disseminated questionnaire included 27 questions and consisted out of four domains: (i) the role of the respondent within the healthcare system; (ii) the current use of reimbursement models, future preferences, and perceived barriers; (iii) the current use of payment models, future preferences and, perceived barriers; and (iv) the experiences the stakeholders have with payment and reimbursement agreements within the specific Dutch policy setting and methods. The questions were asked using Likert scales, rankings, and a few open questions (Questionnaire available in Appendix A). The reimbursement and payment models and their taxonomy included in this study were predominantly based on existing literature with minor adjustments to reflect the Dutch setting (Table 1) [7,19,20,23,27]. Specific definitions are reported in the Appendix A. The list was discussed among the authors with practical experience in the Dutch payment and reimbursement field.

Prior to designing the survey and determining the specific questions asked, exploratory conversations with representatives (N = 6) of the different stakeholder groups were held. Based on these conversations and internal discussions the final questionnaire was constructed. To test for validation and reliability of the survey the developed survey was pilot tested by a number of experts with knowledge of the topic at hand or experience with designing questionnaires to verify the clarity, length, format, and usability of the questionnaire [24,43,44]. These results were not used in the final analysis of the results. The survey instrument was programmed in Lime Survey (LimeSurvey GmBH, Hamburg, Germany) [45].

### 2.3. Survey Dissemination and Analysis

The survey was disseminated between May and July 2021. To secure that all participants had the same taxonomy and definition of the included reimbursement and payment models in mind when filling out the questionnaire, a definition list of the different models was included. Additionally, a knowledge clip was presented to the respondents at the introduction of the questionnaire (Appendix A). Additionally, respondents were made aware that the scope of this study was limited to high-priced hospital drugs. To increase the response rate an announcement and three reminders were sent.

All finalized questionnaires were used to generate the results, even when a few questions were left open. The nature of the collected information was both qualitative and quantitative. Data from quantifiable questions were analysed using Microsoft Excel (Microsoft, Redmond, WA, USA) [46] to calculate measures of frequency and tendency. The results from the open questions were analysed qualitatively using Nvivo 12 Pro (QRS International, Burlington, MA, USA) [47]. Where the answers were analysed using inductive coding.

## 3. Results

### 3.1. Sample Characteristics

Out of the 100 invited participants, 39 (39%) finalized the survey. The healthcare payer perspective was obtained most often (28%, N = 11), followed by a healthcare provider perspective (15%, N = 6), HTA perspective (13%, N = 5), an academic perspective (13%, N = 5), the interest groups perspective (13%, N = 5), a pharmaceutical industry perspective (8%, N = 3), a healthcare consultant perspective (8%, N = 3), and a ministerial perspective (5%, N = 2).

### 3.2. Current Use of Reimbursement Models

In Figure 1, it is shown that financial-based reimbursement models are currently most often applied, where especially discounts were indicated to be applied ‘always’ (29%, N = 11). Respondents indicated that outcome-based reimbursement mechanisms are currently applied less often compared to financial-based models, where only value-based pricing was experienced to be applied ‘often’ (29%, N = 11).

The currently most applied reimbursement model is also the model that is currently best perceived by the respondents. Figure 1 shows that around half of the stakeholders ranked to have the best experiences with discounts (51%, N = 19). The main reason the respondents provided why discounts are perceived so well was due to the simplicity of their implementation. Furthermore, a low administrative burden and the opportunity that discounts provide to get better prices than manufacturers’ list prices were mentioned as prime advantages of this reimbursement model.

### 3.3. Future Preferences and Barriers of Reimbursement Models

In the future, the preferences of the respondents went out to implementing outcome-based reimbursement models more often. Pay-for-outcome reimbursement models were preferred most with 74% (N = 29) of the stakeholders indicating to prefer to apply this model more often, followed by value-based pricing (49%, N = 19), coverage with evidence development (49%, N = 19), and conditional treatment continuation (33%, N = 13).

Given that the most preferred reimbursement model for future implementation is the pay-for-outcome model, a closer look was given at which barriers are currently perceived with implementing this model. Figure 2 shows that according to the respondents, the most perceived barrier was related to difficulties with generating evidence in practice (66%, N = 19). Additionally, high administrative burden (59%, N = 17) and time-consuming negotiations (41%, N = 12) were frequently mentioned.

### 3.4. Current Use of Payment Models

In the experience of the respondents upfront payment is currently the most frequently implemented payment model in the Netherlands with 42% (N = 16) of the respondents indicating that it is currently often to always applied (Figure 3). This stands in contrast to the delayed payment models, which are currently in the experiences of most respondents, rarely to never applied.

The best-experienced payment model is also the most applied one, similar to the findings for reimbursement models (Figure 3). More than half of the respondents (55%, N = 16/29) indicated that upfront payments are perceived best. The most mentioned advantages of upfront payments are related to the simplicity of the implementation and that it often is an acceptable mechanism for the parties involved.

### 3.5. Future Preferences and Barriers of Payment Models

A clear preference is seen for implementing delayed payment models in the future. A total of 62% (N = 24) of the stakeholders indicated a preference for pay-at-outcome achieved payment models to be applied more often in the future (Figure 4). Thereafter, respondents indicated that they preferred annuity payments most (31%, N = 12), followed by health leasing payment models (23%, N = 9).

In Figure 4 the barriers are shown that are perceived with implementing payments at outcomes achieved models more often. Barriers with this most preferred payment model are mostly related to the administrative burden of organizing the implementation (70%, N = 16) and the difficulties with defining outcome criteria to which the payment should be related (52%, N = 12).

### 3.6. Reimbursement and Payment Models within the Dutch Policy Setting

In the current Dutch reimbursement setting the most indicated main objectives, public stakeholders have in mind when making payment and reimbursement agreements were to reduce the price so that the budget impact remains limited or is reduced (68%, N = 24), to control financial risks and uncertainties (68%, N = 24), and to improve access to high priced treatments (63%, N = 24). Most difficulties were perceived with making reimbursement and payment agreements for a combination of therapies, followed by orphan-designated therapies and advanced therapy medicinal products (ATMPs).

Most of the stakeholders (45%, N = 17/38) indicated to be ‘moderately dissatisfied’ (score 2 on a five-point Likert scale where 1 is very dissatisfied and 5 is very satisfied) with how reimbursement and payment models for high-priced hospital drugs are currently organized in the Netherlands, especially the untransparent and confusing system with complex and time-consuming procedures, including the ‘waiting lock’ procedure [46]. Nevertheless, most respondents (58%, N = 19/33) indicated to be ‘moderately satisfied’ (score 4 on a five-point Likert scale where 1 is very dissatisfied and 5 is very satisfied) with the current role their organisation plays within the Dutch policy regarding payment and reimbursement models. However, the respondents argue that attention should be paid to the degree of understanding of each other’s interests and how to best form and agree upon clear preconditions for payment and reimbursement models.

## 4. Discussion

This study showed that for both reimbursement and payment models, the currently most applied models, financial-based reimbursement and upfront payment models, are also the best-perceived ones. Nonetheless, preferences go out to implementing more innovative reimbursement and payment models, especially pay-for-outcome reimbursement models and payment at outcome achieved models, in the future. Nevertheless, there are still multiple barriers hindering a more frequent implementation of the preferred reimbursement and payment models. Specifically, difficulties in generating evidence, high administrative burdens, and time-consuming negotiations, determining the exact outcome criteria to which the payment should be related to and the lack of capacity to implement the scheme in the desired way were perceived as main barriers.

Similar results can be found in previously reported international findings. Several studies concluded that financial-based reimbursement and upfront payment models are currently implemented more frequently than outcome-based reimbursement models and delayed payments, respectively [7,21,27,41,48,49,50,51], as also indicated in cross-country comparisons [17,24,27,36,50]. Outcome-based reimbursement models and delayed payments models are seen as promising alternatives to reduce remaining (clinical) uncertainties, diminish the budget impact, share financial risks, and improve/ensure patient access [21,52,53]. Nonetheless, it is still debated how much is and can be gained with the implementation of outcome-based reimbursement and delayed payment models due to their complexity and administrative burden [27,28,29,30,31,32,54]. The perceived barriers stakeholders expressed when implementing the preferred outcome-based reimbursement and delayed payment models clarify why the preferred models are currently not yet commonly applied in the Netherlands. A review by Makady et al. outlined the difficulties of implementing conditional financing as a form of an MEA in the Netherlands in the desired way where it was concluded that due to procedural, methodological, and decision-making issues the reimbursement model did not meet its aims or only partially did so [25]. Correspondingly, the main reason why discounts and upfront payments are experienced best according to the respondents, namely the simplicity of these agreements, can be explained. Some of the described barriers perceived were also more broadly experienced across Europe and the United States [7,16,27,28,31,32,36,52,54]. Bohm et al. presented a clear overview of the main challenges related to successfully implementing outcome-based agreements across Europe, namely the lack of useful negotiation frameworks; difficulties in determining outcomes; the burden of data administration and implementation; and laws and regulation [22]. Additionally, Michelsen et al. pointed out similar barriers, emphasizing how delayed payments are difficult to arrange due to 12-month budget cycles and how outcome correction of payments is currently hindered by the need for additional data collection, the lack of clear governance structures, and the resulting administrative burden and cost [21]. These results show that many countries are still experiencing a broad range of barriers when implementing more innovative reimbursement and payment models. Nonetheless, European healthcare payers do seem to have a positive attitude towards innovative reimbursement and payment agreements, and their implementation is expected to increase in European markets [17]. Especially, the expected increase of advanced and regenerative medicines, such as cell and gene therapies, with substantial upfront costs and large clinical uncertainties at the time of reimbursement and payment decision-making calls for such new models. Consequently, more and more opportunities are seen in exploring how delayed payments and outcome-based reimbursement agreements can be based on the actual value received by patients and which role real-world data could play, with the use of advanced analytics, including artificial intelligence and machine learning, in the move from conventional payment and reimbursement models to more innovative managed entry agreements [6,10,16,21,22,24,29,52,55].

### 4.1. Implications and Directions for Future Research

Given that multiple respondents indicated to be missing a clear overview of the payment and reimbursement system, future research should focus on creating a clear framework that can be applied in country-specific settings to ease and fasten the process in the future. Additionally, the consensus among the different stakeholders should be reached about what the criteria are to successfully implement more innovative reimbursement and payment models, e.g., how to set clear outcomes to which the payment should relate, what the minimum data infrastructure is that should be in place and how the administrative burdens will be shared. The development of a negotiation framework, checklist templates, and a calculation tool to outline the effects of using different models are examples of what can be done to ease the process of negotiating more innovative reimbursement and payment agreements [11,56]. Additionally, it should be encouraged to perform more pilots as well as systematic evaluations of established innovative payment and reimbursement agreements [33]. International collaboration may enhance the transparency surrounding implemented agreements and benefit-sharing best practices to support further payment and reimbursement decision-making. Finally, the studied reimbursement and payment models are not mutually exclusive and often combined into one agreement when implemented. Future research should focus on exploring which combinations of reimbursement and payment models are most suitable to address different issues within the same agreement (e.g., budget impact and use, access, and cost-effectiveness) where recommendations should be made fit-for-purpose for specific national settings [21,23,57,58].

### 4.2. Limitations

The survey was targeted at stakeholders who are directly or indirectly involved with payment and reimbursement in the Netherlands. However, it was not possible to contact all relevant stakeholders involved within each stakeholder group and not all invited stakeholders responded to the survey. This emphasizes that caution should be taken with generalizing the results to the entire Dutch reimbursement and payment setting. Nevertheless, a high level of homogeneity was found in our outcomes, and we aimed to invite key stakeholders based on their seniority, with a vast knowledge of their fields within the Dutch payment and reimbursement setting and were therefore deemed adequate to provide a sound representation regarding the experiences and preference. Finally, this study sheds light on the current use, future preferences, and perceived barriers of reimbursement and payment models within the specific Dutch setting, therefore the results and its conclusions will not always directly apply to other countries.

## 5. Conclusions

Outcome-based reimbursement models and delayed payment models are preferred to be applied more frequently in the future. Financial-based reimbursement models and upfront payment models are currently implemented most and are also perceived best by the different stakeholder groups in the Netherlands, mainly due to their simplicity. Barriers perceived with the preferred outcome-based reimbursement and delayed payment models mostly related to evidence generation and administrative difficulties may complicate their implementation. Future efforts should offer stakeholders a decision and implementation framework specific to the national setting.

## Figures and Tables

**Figure 1 ijerph-20-00340-f001:**
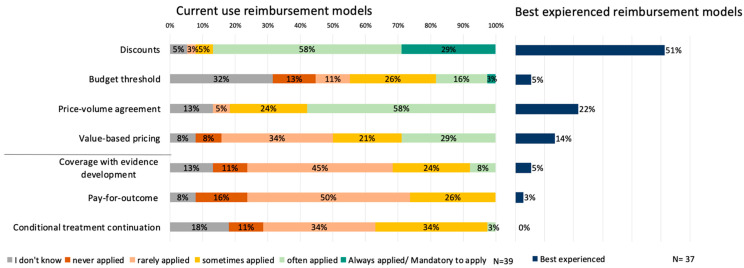
Current use of (**left**) and best experienced (**right**) reimbursement models.

**Figure 2 ijerph-20-00340-f002:**
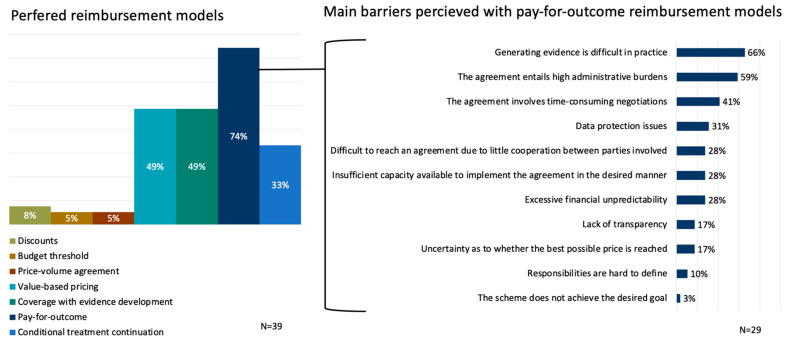
Preferred reimbursement models (**left**) and the main perceived barriers with pay-for-outcome models as the most preferred model (**right**).

**Figure 3 ijerph-20-00340-f003:**
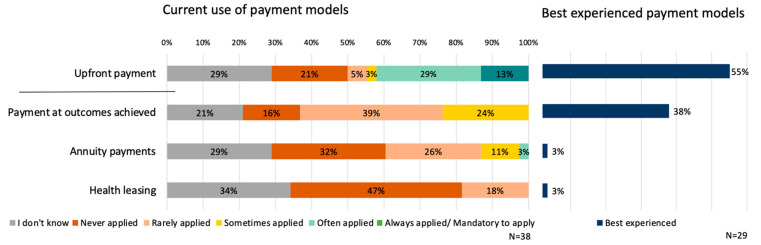
Current use of (**left**) and best experienced (**right**) payment models.

**Figure 4 ijerph-20-00340-f004:**
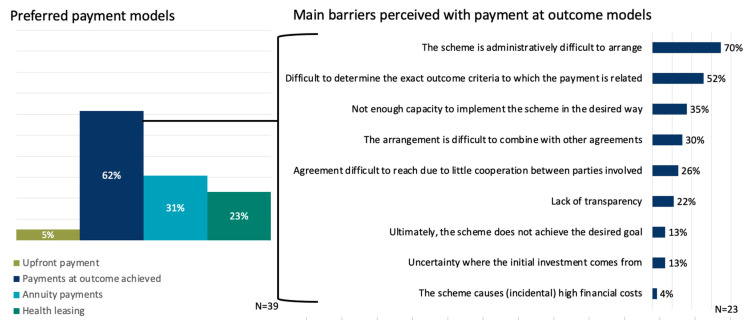
Preferred payment models (**left**) and the main perceived barriers with payment at outcomes achieved models as the most preferred model (**right**).

**Table 1 ijerph-20-00340-t001:** Included reimbursement and payment models.

Types of Models
*Reimbursement models* Financial-based reimbursement models Discounts/rebates Budget threshold/dedicated funds Price-volume agreements Outcome-based reimbursement models Value-based pricing Pay-for-outcome/outcome guarantees Conditional treatment continuation Coverage with evidence development
*Payment models* Upfront payment Delayed payment models Pay at outcomes achieved Annuity payments Health leasing/subscription

## Data Availability

Not applicable.

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
