# Peer review of "When Reality Does Not Meet Expectations—Experiences and Perceived Attitudes of Dutch Stakeholders Regarding Payment and Reimbursement Models for High-Priced Hospital Drugs"

_ijerph, 2022, doi:10.3390/ijerph20010340_

Round 1

Reviewer 1 Report

Thank you for giving me the opportunity to read and comment a report “When reality does not meet expectations. Experiences and perceived attitudes of Dutch stakeholders regarding Payment and Reimbursement models for high-priced hospital drugs”, by Callenbach M.H.E, et al.

In the reviewed manuscript, the Dutch stackholders´ experiences and expectations with payment and reimbursement mechanisms has been analyzed. 

This paper is well written, correctly structured with a suitable research concept, the study limitations are addressed, and it is of relevance to readers of the journal. However, I include a few comments for your consideration.

·       To validate the internal consistency of the questionnaire based on the Likert scale, the Cronbach's Alpha should be calculated and interpreted.

·       The percentage of participants who completed the survey is not very high. This should be mentioned in the discussion.

Author Response

Dear reviewer,

Many thanks for taking the time to review our manuscript and providing us with valuable feedback. In the below bullet points, we describe how we have addressed your comments:

  • To validate the internal consistency of the questionnaire based on the Likert scale, the Cronbach's Alpha should be calculated and interpreted.
    • Response: We have calculated the Cronbach's Alpha for both Likert scale questions, which were 0,75 and 0,9 respectively. These values can be interpreted as acceptable and excellent. We are confident that this supports the international constancy of the questionnaire based on the Likert scales. However, we are unsure how to further incorporate/interpret the internal consistency for these Likert scale questions as they questioned how often certain types of payment and reimbursement models are used.
  • The percentage of participants who completed the survey is not very high. This should be mentioned in the discussion
    • Response: Thank you for pointing this out. We have highlighted this in the limitation section of the discussion. Given that we invited a (very) broad group it was highly possible that stakeholders were invited that felt not completely able to answer the quite detailed questionnaire. We, therefore, considered the response (39%) as relatively high. As a comparison, the average response rate of online surveys is 33% according to Wu, M. J., Zhao, K., & Fils-Aime, F. (Response rates of online surveys in published research: A meta-analysis. Computers in Human Behavior Reports 2022, 100206)

We hope that through these elaborations we have sufficiently addressed your comments.

With kind regards,

Marcelien Callenbach

Reviewer 2 Report

The manuscript was relatively well written and can be considered for publication after minor revision as suggested below. 

First, a differential analysis of gender can be included in the analysis of the sample. In the extant literature, gender is a key influencing factor.

Second, it is suggested to add some comparisons of relevant studies from other countries.

Author Response

Dear reviewer,

Many thanks for taking the time to review our manuscript and providing us with valuable feedback. In the below bullet points, we describe how we have addressed your comments:

  • First, a differential analysis of gender can be included in the analysis of the sample. In the extant literature, gender is a key influencing factor.
    • Response: Thank you for pointing this out, however, a differential analysis of gender is not possible since gender was intentionally not included in the survey questions. This was not deemed a relevant factor in the analysis of how stakeholders (from the perspective of the institution/organization they work at) identify the current use of and preferences for payment and reimbursement models.
  • Second, it is suggested to add some comparisons of relevant studies from other countries.
    • Response: Thank you for making this suggestion. We feel that the comparison of relevant studies from other countries is quite complete (line numbers 269 to 289), however, if you have suggestions of relevant studies that we missed we would be glad to include them.

We hope that through these elaborations we have sufficiently addressed your comments.

With kind regards,

Marcelien Callenbach